

# Frameworks for developing an Agro-Prosumer Community Group platform

Pratima Jain[1] and Vidyasagr Potdar[2]

[1] School of Information Systems, Curtin University of Technology, Perth, WA, Australia
[2] School of Management, Curtin University of Technology, Perth, WA, Australia

## ABSTRACT

**Background**. The involvement of prosumers in the form of agricultural community groups has been acknowledged, and interest in it is increasing due to local food demand and quality of food. How to create a prosumer group? The definition of agro-prosumers and analysis of their behaviour, engaging new members to the existing groups, managing members and their goals are important factors to consider. Hence, to overcome this barrier and to improve the participation of prosumers, in this paper three key frameworks are presented to develop an Agro-Prosumer Community Group (APCGs) platform.

**Methods**. A conceptual process that consist of strict multiple stages i.e., requirement analysis, design logic, theoretical fundamentals, implementation of prototype and verification, is used to build the frameworks for APCG. Different methods and approaches are used to design and develop framework's prototype. For instance, clustering algorithms are used to define and group agro-prosumer concept, an approach is developed that evaluates real-time production behaviour of new prosumers while engaging them to APCG. Finally, the goal-ranking techniques i.e., MCGP are used to build a goal management framework that effectively reaches a compromise between diverse goals of APCGs.

**Results**. Results for each framework is shown while verifying the prototype using prosumers data.

**Conclusion**. An Agro-Prosumer Community Group addresses three key issues relevant to the development of an agro-prosumer community-based approach to manage the prosumers in local food- and carbon-sharing networks. The key contributions are (1) APCG concept, (2) Prosumer engagement framework, and (3) Goal management framework. Thus APCG platform provides a seamless structure for carbon and produce sharing network.

## INTRODUCTION

Prosuming, individually or as a group, is seen as a political act which is feasible, reduces unfavorable environmental changes and affects the economy by reducing centralized long-chain value in the supply chain (*Jessen-Hannula, 2019*). Thus, prosuming in urban agriculture can be perceived as a sustainable step for the agriculture industry.

Corresponding author
Pratima Jain,
Pratima.jain@postgrad.curtin.edu.au,
au.pratima@gmail.com

With the growing interest in the supply of quality food, the urban prosumer can be seen as a leader in providing high quality, trustworthy produce. Prosumers' crop yields are not only considered high quality as generally they are home grown, but are also chemical free, grown in better soil (*i.e.,* composting) or grown organically. Additionally, prosumer crops are more trustworthy and better quality than commercial produce as they are usually home grown from seeds or seedlings, as is the practice of the urban prosumer. Hence, an agro-prosumer has strong prospects of obtaining better value and exposure in the market by forming or joining a prosumer community group.

An agro-prosumer community group (APCG) can be described as a community group network formed by "using different agro-prosumers profile, personal motivations and unique characteristics". Forming an agro-prosumer community group has a number of benefits such as it can improve economic value and offer rich socio-psychological experiences (*Jessen-Hannula, 2019*) to the agro-prosumers by creating social relationships and self-pride, imparting new skills to members, generating knowledge, and contributing to community activities. In addition to high quality produce, agro-prosumers can generate carbon tokens depending upon the consumption of the total amount of carbon content consumed during the vegetation process, and trade it with industries. APCGs can also reduce the long supply chain work, thus improving transparency, security and sustainability.

To build an APCG network, the first step is to define APCG and identify prerequisites. To achieve this, a framework is designed where agro-prosumers' profiles are assessed and analyzed to form different groups and derive each group's unique pre-requirements. These unique requirements will become the prerequisites of each group and will be utilized to classify prosumers into appropriate group. The framework utilizes agro-prosumers' production history when deciding the pre-condition criteria for each APCG.

Furthermore, engaging new agro-prosumers is critical to make APCGs a sustainable network. Thus as a next step an APCG require a recruitment framework to add new agro-prosumer in the network. For new recruitments, it is important to evaluate the new agro-prosumers' real-time production profiles before offering them membership of their desired APCG. Thus, rather than relying on historic production behavior, it is important to use real-time production profiles, which will give a better understanding of the prosumer's commitment to supporting the APCG. Hence, we propose a framework for recruiting new agro-prosumers for an APCG. The recruitment is based on an evaluation of their real-time commitment conducted over a defined period.

After initiating the community-based, produce-sharing network, one of the key requirements is to make APCGs goal-oriented. This can be done by determining the overall community objectives of the production-sharing network and, subsequently, managing diverse multiple goals of the various APCG groups.

Goal management can be challenging in a community-based network due to diverse conflicting issues such as demand constraints and cost constraints. Several situations can occur where one objective is achieved while leaving another. For instance, in order to improve carbon sequestration in soil by APCGs, organic/ecological farming ways must be practiced; however, organic farming methods yield less, which in turn affects the collective produce of the APCGs and subsequent income. Therefore, it is a requirement that a

compromise be applied to the multiple goals with respect to the given constraints. Based on the above discussed factors, an effective framework for the management of goals is essential. Thus, another key framework is developed and termed as goal management framework. The paper presents a seamless structure to develop an agro-prosumer community group network by proposing and verifying three sub frameworks *i.e.,* APCG definition and prerequisites, APCG new prosumer recruitment and goal management for APCGs.

# MATERIALS & METHODS

## Framework 1 APCG definition and prerequisites

The key input for the APCG's definition and pre-conditions-determining framework is the agro-prosumer's produce profile. Agro-prosumers' produce profiles are selected based on the suburb or postcode and types of crops grown in their garden, along with the quantity during different seasons, particularly winters and summers. Outputs derived from this framework will become the prerequisites for different APCGs. The pre-condition requirement for each member will be treated as a commitment to meet their group's prerequisites.

The framework is divided into two parts as shown in Fig. 1. Phase 1: clustering prosumer profiles and outlier detection, and Phase 2: optimizing prosumer clusters to define group's pre-conditions.

Agro-prosumers' seasonal summer and winter data has been collected as an input for the first part of the framework. Kmeans algorithm has been worked out, however the objective here is to find out prosumers based on similar production behavior. Therefore, an hierarchical clustering algorithm shown in Fig. 2 is used to create clusters based on the homogeneity of prosumers' profiles, and to detect outliers. Homogeneity of prosumers with similar produce profiles will help in earning fair amount of incentives for all and will also support easy incentive distribution (*Rathnayaka et al., 2015a*).

Clusters are optimized and unique attributes are identified in the second part of the solution. The non-overlapping agro-prosumer clusters from the first part are optimized to achieve a feasible number of APCGs, and unique attributes are identified for each group and used as pre-requisites of APCGs.

### Phase 1 Prosumer clustering

The first phase includes clustering of the prosumers' profiles and detecting any outliers using a hierarchical clustering method. Prosumers' seasonal profiles for two seasons (summer and winter) are taken as an input for this phase. There are three steps in this phase: creation of regional groups, building clusters, and outlier detection.

### Step I Creating regional groups

In this phase, agro-prosumer's postcode is taken as an input. The prosumers are partitioned into groups based on their postcodes within a certain region. This will mean that the delivery of prosumer produce can be done without the need for long-distance transportation. The output of this step will provide GL-clusters (geographical location based-clusters) based on postcodes and the neighborhood zone.

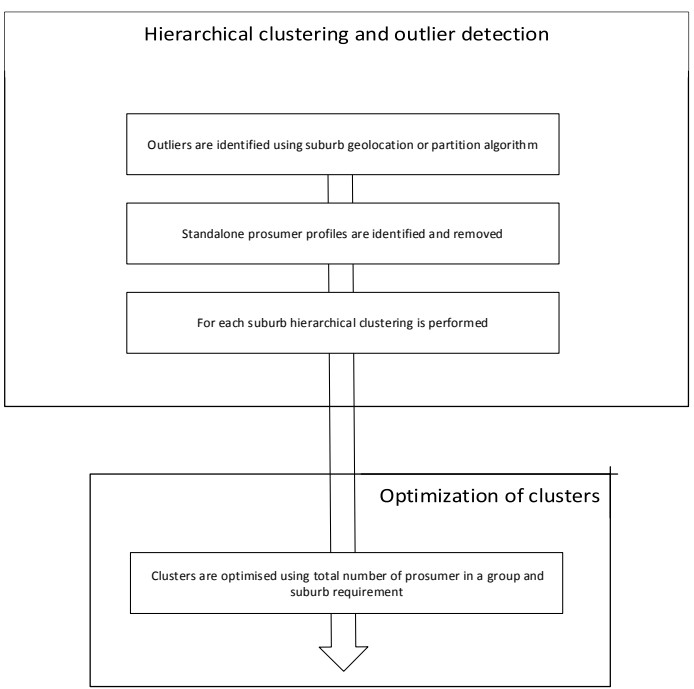

**Figure 1 Theoretical foundation of APCG definition and prerequisites.**

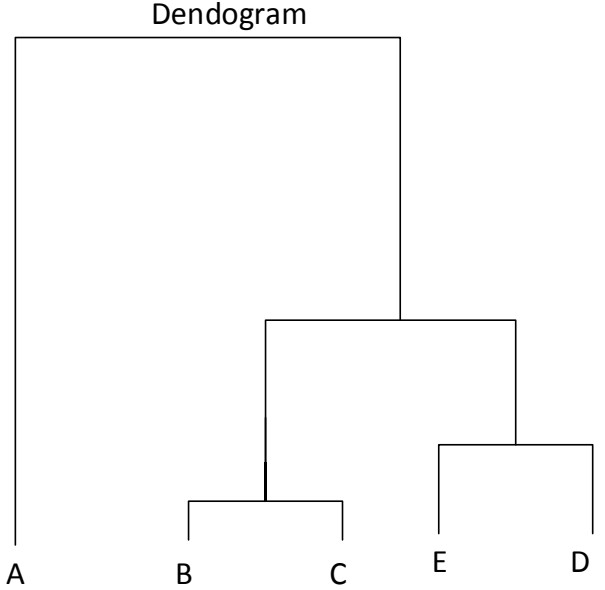

**Figure 2 Hierarchical clustering.**

### Step II Outlier detection

In order to deal with outliers, a threshold is set: after calculating the distance between existing clusters, if the shortest distance is not further than the threshold, we assign the dataset to its closest cluster. If the shortest distance exceeds the threshold, this means that the cluster could belong to a minor group. The objects in the minor group are those that did not belong to any major groups. Objects in minor groups are data points, not outliers as they do not belong to any major groups. Further clustering of objects in minor groups can be done for future analysis.

### Step III Building clusters

For each GL-cluster obtained from step one and after removing outliers, the corresponding agro-prosumer profiles are considered in the next step, and clusters are formed based upon prosumers' production history. The hierarchical clustering method is used to decide the number of clusters. Initially, each prosumer profile is placed in a unique cluster. For each pair of clusters, some value of dissimilarity or distance is computed. In this case, minimum variance, *i.e.,* Ward's criterion, is used to minimize the total within-cluster variance and find the pair of clusters that leads to minimum increase in total within-cluster variance after merging. In every step, the clusters with the minimum variance in the current clustering are merged until the whole dataset forms a single cluster. Hierarchical clustering helps in identifying groups in the dataset. Thus, the output from this step will be number of prosumer clusters based on their production similarity.

### Phase 2 Prosumer cluster optimization and forming pre-requisites:

This phase involves the optimization of prosumer groups based on the number of prosumers in each group and their production amount. Firstly the clusters are optimized and pre-requisites for each cluster-group is formed. The optimization steps and pre-requisites are further illustrated in this section.

### Step I Optimization of prosumer clusters

Agro-prosumer cluster-groups created by using hierarchical clustering are optimized to produce sufficient number of clusters that will then represent different agro-prosumer community groups. The number of clusters produced by optimization, depends on the variation of production quantity. If the variation is large, too many clusters could be formed, which are not feasible to manage. Thus, this stage involves optimizing the clusters into a feasible number of APCGs by merging small clusters into one or splitting large clusters into smaller ones to obtain a feasible number of APCGs to satisfy market requirements. In order to determine the ideal number of clusters, firstly, suburb requirements are analyzed and the expectations of relevant APCGs are derived.

To optimize APCGs:

Let X be the population of suburb ABC and C is the per capita consumption of lemons. Assume that the APCG framework targets a minimum 1% of lemon market for a suburb ABC. Then the requirement ($R_{expected}$) of lemons for suburb ABC using APCGs can be calculated with

$R_{expected} = X^{\star}C^{\star}0.01.$

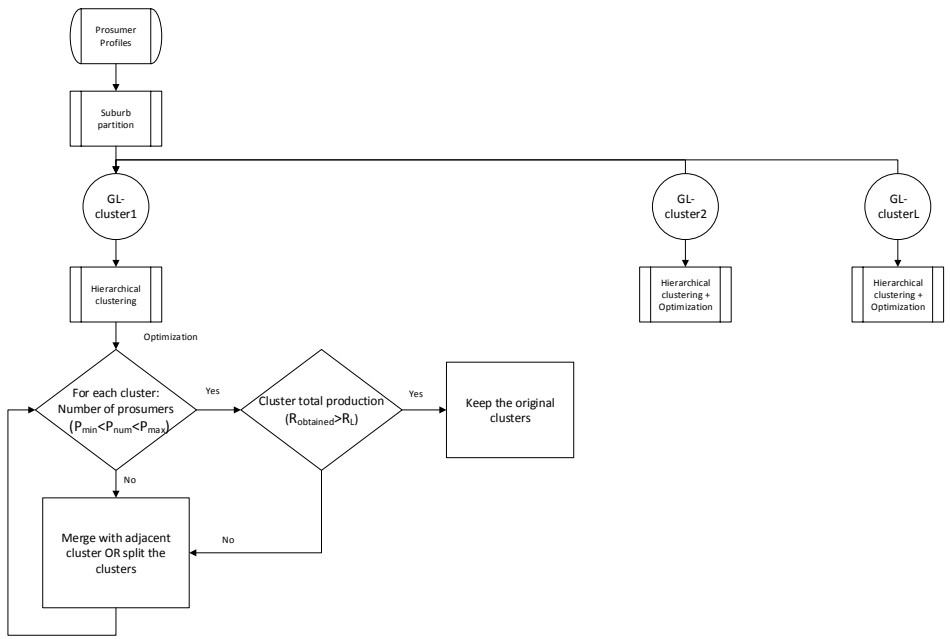

**Figure 3  Flowchart for Hierarchical and Optimization of clusters.**

Let L be the number of clusters formed using the clustering method, and $R_L$ represents every APCG's goal.

$R_L = R_{expected}/L$.

After determining the suburb's requirements, next step optimizes the clusters by evaluating number of agro-prosumers present in each APCG (as shown in Fig. 3). Let say $P_l$ and $P_h$ respectively be the lowest and highest number of prosumers expected in each group. Let $R_L$ be the minimum amount of production expected from each APCG. Prosumers count ($P_{num}$) and the production quantity ($R_{obtained}$) from a specific prosumer cluster is shown in Eqs. (1) and (2).

$$P_l < P_{num} < P_h \tag{1}$$

$$R_{obtained} > R_L \tag{2}$$

If the production is less than the expected amount ($R_{obtained}<R_L$), or the number of prosumers is lower than the ideal number of prosumers ($P_l>P_{num}$), that agro-prosumer cluster is merged with the closest prosumer cluster, and the same process continues until the prosumer cluster can meet the total production requirement ($R_{obtained}>R_L$) and the number of prosumers ($P_l<P_{num}<P_h$) defined for the APCG.

Now, if too many prosumers form an agro-prosumer clusters, the clusters are further break down into small size clusters consisting of an most favourable number of prosumers and meeting the production goals *i.e.,* $R_{obtained}>R_L$ and $P_l<P_{num}<P_h$.

The final output of optimization will result in the optimised prosumer clusters, which are then represented as APCGs. Now these APCGs are analysed to identify the unique production characteristics or pattern of each group which will be denoted as the pre-requisite of the APCGs.

### Pre-requisite formation

Introduction to APCGs includes formation of unique entry requirements for each group. The two key input, as discussed in the previous sections, to determine the prosumers' adherence are the "lower threshold" ($Lt$) and the "upper threshold" ($Ut$). The defined inputs used as pre-requisites of each APCGs will be:

- Lower threshold: $L_t$
- Upper threshold: $U_t$

### Framework 2: agro-prosumers recruitment framework

A new recruitment framework is designed to evaluate real time behavior of new agro-prosumers and allocate them in specific APCG groups. The reason for designing this approach is to encourage participation of non-farmers and new gardeners, which not only help them to estimate production details, outline incentive benefits etc., but will also ease off the management of APCG in the long run. Additionally this framework requirement is new and won't be justified to use CSA methods which basically works on partnership basis and useful for a large piece of land (*Ahluwalia & Miller, 2014*; *Cone & Myhre, 2000*; *Blattel-Mink et al., 2017*). An overview of the framework is shown in Fig. 4.

New agro-prosumers who are interested in joining the APCGs, and their real-time behavior profiles, are collected as input for this framework. We term these agro-prosumers "prospect agro-prosumers" who are assumed to be new to the community sharing network; thus, because there are no previous production profiles, real-time production needs to be determined. The final outcome of this framework is the recruitment of prospect agro-prosumer to suitable APCGs. This stage is further divided into four components, which are explained below.

The framework has four components:

1. An approach to evaluate agro-prosumers' production performance;
2. Agro-prosumers' transaction assessment during the evaluation period
3. An approach to analyse agro-prosumers' stability; and
4. Agro-prosumers recruitment to a specific APCG after the evaluation period.

The varying nature of agro-prosumers' production behaviour is evaluated using the above approaches, and allocates them to a temporary "variable APCG". Later on, the prosumers' overall behaviour is stored and evaluated prior to recruitment to a specific APCG, *i.e.,* to one of the final APCGs. The requirements for the proposed solution are covered *via* four components (listed above) discussed in detail below.

### Approach to evaluate prosumers' production performance

Finding an approach to estimate agro-prosumers' performance is the first component of the evaluation technique, which helps in understanding the evaluation period activities and evaluation inputs.

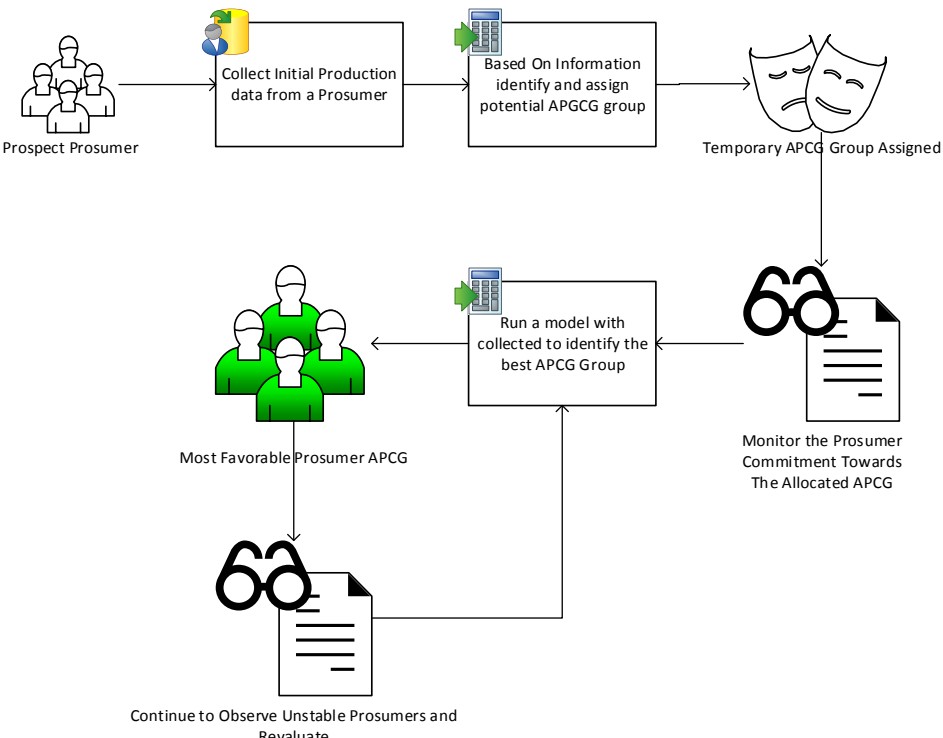

**Figure 4** **Overview of agro-prosumer recruitment framework.**

## Agro-prosumer evaluation measures

As discussed previously, the "evaluation period" is an established period of consecutive seasons during which the production behavior of new agro-prosumers who are interested in joining an APCG, is evaluated. The evaluation period is divided into two seasons per year in Australia: winter (*i.e.,* March–August) and summer (*i.e.,* September–February). These winter and summer seasons show non-overlapping, mutually exclusive time periods and are assigned with a production transaction between agro-prosumer and the APCG module using production value.

Agro-prosumers' production data is generated using the Australian national average. Production data such as family size, farming methods (organic, inorganic), lemon variety (three major lemon variety has been used) and number of trees (1–10 has been randomly used) and their respective ages (age of a tree is assumed from 5–100 years), are collected as input to evaluate their consumption pattern and production performance for two season or annually. Agro-prosumers' surplus production is considered as the final value for one season/year. Thus, prosumers' performance is estimated using that final value, and is evaluated for each season. Next section explains the approach used to determine the prosumers' performance for each season during the evaluation period.

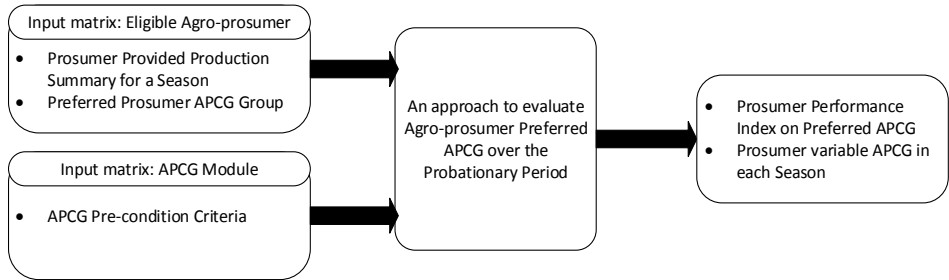

**Figure 5** Approach overview to evaluate agro-prosumer.

**Table 1** Performance scale interpretation.

| Performance index interpretation | Success/failure rate | Performance score |
| --- | --- | --- |
| Complete success | 100% | 3 |
| Intermediate | 90–99% | 2 |
| Entry | 80–89% | 1 |
| Failure | 0–79% | 0 |

### The proposed approach

This approach requires two inputs: the input from the agro-prosumer and the input from the APCG module as shown in Fig. 5. Inputs from the agro-prosumer include production summary for a season and the prosumer's preferred APCG. The APCG module's input comprises the pre-requisites of the available APCGs.

A probabilistic approach is used here to evaluate agro-prosumers' production performance based on the pre-condition criteria of their preferred APCGs.

Results of this approach are the "performance indices" and variable APCG of the agro-prosumer for each season. Performance indices are used to anticipate the level of success and/or failure of an agro-prosumer in meeting the pre-condition criteria of his/her preferred APCGs. To utilize it, different levels of success and failure are represented using a four-point scale as shown in Table 1. In fact, each performance index shows different value or success rate of performance in the production behavior.

The performance scale ranges from 0 to 3, where 3 represents the complete success or match, and the minimum success rate is 80% for meeting the pre-condition criteria. If the success rate is less than 79%, it will be considered as a "failure".

The performance scale used here has single-integer values. It is difficult to use extreme values, *i.e.,* only high or low, to measure prosumer behavior. Hence, in order to determine and model the performance of prosumers more accurately, various levels of performance should be identified first. Moreover, to accurately determine prosumer performance, the various levels of performance must be identified. A performance score with a value from 0 to 3 will help to indicate prosumers' performance for APCGs development.

• Complete success: The highest point on the performance Score is 3, which indicates "Complete success". This score suggests 100% success rate in interacting with the

prosumers' production-sharing process. This level of performance according to the PS suggests that the prosumer is strongly suited to his preferred APCG and meets the desired pre-condition criteria.

- Intermediate success: This level denotes 90–99% of success rate in interacting with prosumers' production behavior. Performance Score 2 shows that it is the "medium success" level. This score suggests that in meeting the prosumers' preferred APCG requirements, prosumers' performance reliability is good.
- Entry success: Performance score 1 indicates "Entry success". This score suggests 80–89% success rate while satisfying the pre-requirements of the preferred APCG's. This performance index score suggests that the prosumer is slightly reliable in meeting the desired pre-condition criteria of his/her preferred APCG.
- Failure: 0 reflects the lowest score in performance, indicating "failure". This level depicts 0–79% rate of success in fulfilling the pre-requirements. Thus, this level shows that the prosumer's performance is not reliable enough to meet the pre-condition criteria for the APCG. Hence, the prosumer with this index could be matched with other APCG rather than the preferred one.

The mathematical expression of performance indices is given in Eq. (3).

For a season (j) of the evaluation period, the rate of success of the prosumer ($P_{ij}$) being allocated to prefer variable APCG ($C_p$):

$$Rate\left(P_{ij} \in C_p\right) \begin{cases} 100\% : if\ E_{ij} \geq L_p \\ \dfrac{E_{ij}}{L_p} : if\ E_{ij} < L_p \end{cases} \tag{3}$$

where $P_{ij}$ is an i$^{th}$ agro-prosumer's performance in the j$^{th}$ season, $C_p$ is the preferred APCG, $E_{ij}$ is the real time production commitment of i$^{th}$ agro-prosumer and $L_p$ is the production threshold of agro-prosumers preferred APCG.

### Agro-prosumers' transaction assessment during the evaluation period

For ongoing assessment during the evaluation period, agro-prosumer is aimed to assign into his chosen APCG for each season. The evaluation process is shown in Fig. 6. The key steps of the process are as follows: the prospect prosumer is asked to submit records of production in real time for "n" seasons during the evaluation period. For each season, dynamic production amount is compared with the minimum threshold ($E_{th}$), which is the minimum requirement of any APCG. If the prosumers' production is equal or greater than the $E_{th}$, the prosumer is viewed to be an eligible prosumer.

Next, if a prospect agro-prosumer receives 'eligible prosumer' status during his/her first season, she/he will be promoted to the next season and then to following seasons. However, if she/he fails to meet the eligible agro-prosumer requirement, in the first production season, the evaluation period will be extended with more seasons.

However, if the new agro-prosumer is not able to match the minimum threshold ($E_{th}$), then the prosumer's evaluation period is extended by another season and the prosumer remains under evaluation until succeeded. On the completion of the evaluation period,

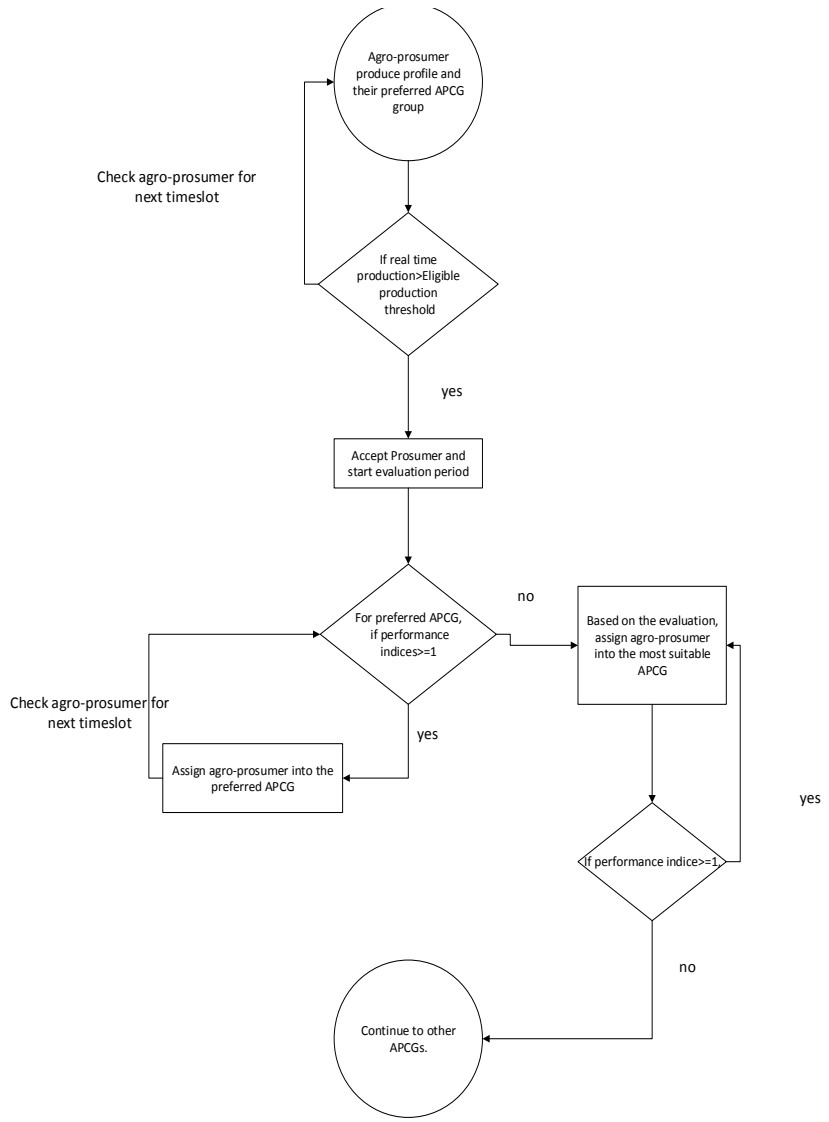

**Figure 6** **New agro-prosumer evaluation process.**

prospect agro-prosumers' stability will be analyzed using stability index, which is discussed next.

### An approach developed to analyze agro-prosumer stability

The stability of an agro-prosumers' reliability is estimated for his/her preferred APCG, as well as for those assigned throughout the evaluation period. Figure 7 shows a process to obtain prosumers' stability for agro-prosumers' chosen APCG.

During evaluation period, for each season, agro-prosumers' performance index values are taken as an input along with their temporary APCGs. Equations (4) and (5) formulate a mathematical equation for the approach. SI represents the stability index which is used to

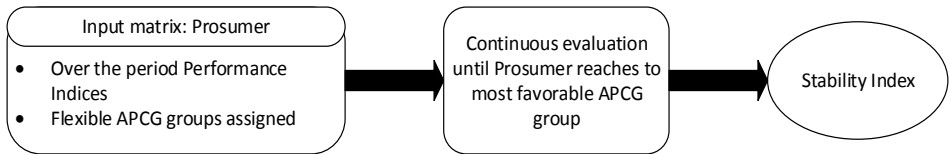

**Figure 7** **Approach to determine stability index for new prosumer in APCG.**

determine the feasibility that prosumers will remain in their preferred APCG. The output for $I$ index is between 0 and 3, and a higher $I$ shows high chances of prosumers remaining in their preferred APCG:

$$I_{pi} = \frac{\sum_{j=1}^{ns} PX_{ij}}{ns}. \tag{4}$$

Above $I_{pi}$ is the stability index of the $i^{th}$ prosumer with respect to chosen APCG ($C_p$), $PX_{ij}$ is an $i^{th}$ prosumers' performance index in the $j^{th}$ season and ns is the number of seasons where the prosumer is assigned to his/her chosen APCG. To determine most suitable APCG for an agro-prosumer, rate of engagement to a specific APCG is calculated using Eq. (5). For example, if the agro-prosumers' rate is higher for APCG1 than other APCGs, than the chosen APCG1 is seen as the most favorable APCG for that prosumer's engagement.

$$Rate(P_i \in APCG_{Fr}) = \frac{count\ of\ (APCG_{Fr})}{ns} \tag{5}$$

where $P_i$ is the $i^{th}$ prosumer, $APCG_{Fr}$ is the $r^{th}$ temporary APCG, count of ($APCG_{Fr}$) shows the total number of times the prosumer is selected to $r^{th}$ temporary APCG during the evaluation period and ns is the number of seasons.

The next section discusses agro-prosumer engagement to the permanent APCG based on the previously-described method.

### Agro-prosumers engagement to the permanent APCG after the valuation period

Agro-prosumer engagement to the most suitable APCG is analyzed in this step. The overall performance of prospect agro-prosumers overall performance is assessed at the end of the evaluation period. Figure 8 is a flowchart showing this process. As discussed in the previous section, the Stability Index, based on an agro-prosumer's performance index, is calculated throughout the evaluation period. Additionally, agro-prosumers' rate of staying in temporary APCGs during the entire evaluation period is assessed. Equation (6) is utilized to identify the combined value of the agro-prosumer being allocated to the permanent APCG. The APCG which shows the highest joined index is chosen as that prosumer's final permanent APCG.

$$IPr(P_i \in APCG_{Fr}) = I_{pi} \times Rate(P_i \in APCG_{Fr}). \tag{6}$$

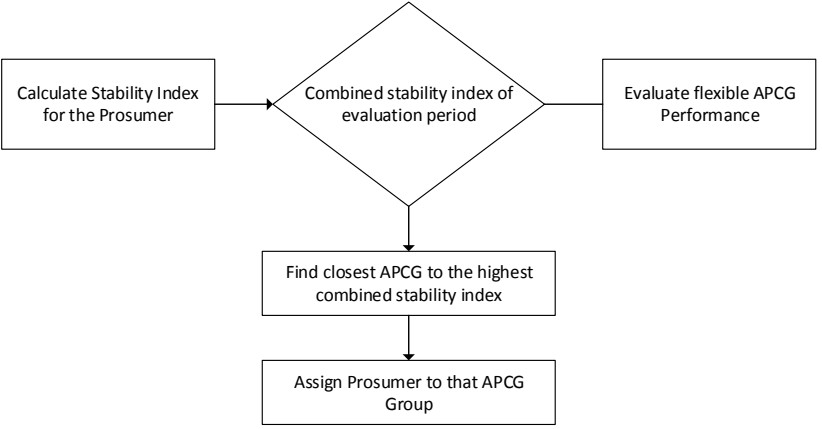

**Figure 8** agro-prosumer recruitment process.

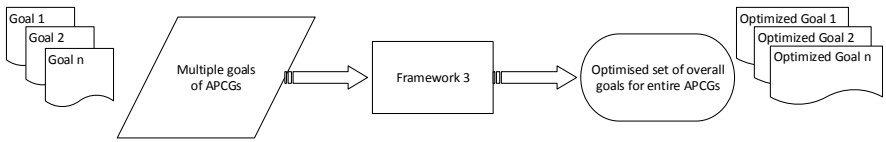

**Figure 9** Concise overview of the goal management framework.

### *Framework 3: Goal management framework*

The input for the framework includes diverse goals for agro-prosumer community groups.

The solution framework consists of a goal management component. The outcome of the goal management phase is an optimized set of overall goals for the community-based, harvest-sharing network. The processes involved in goal management are shown as an overview of the framework in Fig. 9.

### *Goal management*

The goal management stage is responsible to attain ideal goals structure out of overall goals. The purpose involves solving diverse conflicting goals in the APCG to obtain best solution in terms of goals priority. The feature of MCGP (*Ravindran, 2009*) and an approach utilised in smart grid goal management (*Rathnayaka et al., 2015b*) is referred to design best possible solution for conflicting goals. To achieve this, each and every identified objective is attached with a rank based on their priority. High rank objectives are treated as goals to work out first, and therefore attempts are made to find a solution which is close to the pre-ranking set of goals. Goal programming minimises the deviation between the theoretical goals and realistic achievements. These deviation can be both positive and negative, thus an objective function is used to minimise the deviations based on the relative importance of the goals.

Various areas has utilised goal programming model benefits such as environment, energy, smart grid, academic and health planning (*Rathnayaka et al., 2015a*), and shows

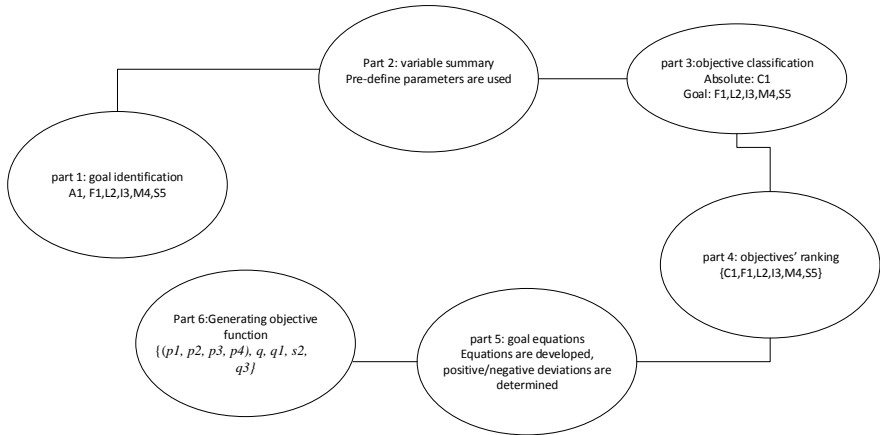

**Figure 10** **Goal programming model.**

success in solving diverse conflicting goals. In this framework, we adapt MCGP techniques for our framework. Figure 10 presents the algorithm for the goal programming model, where the parameters and equations are explained in the following section. The model has six parts:

  (i)  APCGs goal recognition,
 (ii)  Summary of variables,
(iii)  Objective classification,
(iV)  Objective ranking,
 (v)  Goal equation formation, and
(vi)  Generating objective functions.

### Part 1: APCGs goal recognition

APCGs diverse goals are identified in this phase. These objectives are explained below.

  I. Carbon content objective (C1): The "carbon-capture objective" refers to the use of organic farming methods to maximize carbon capture, which will increase the carbon content which can be traded with external companies. More carbon capture will result in more carbon sequestration and less emission.

 II. Food security within the network (F1): The goal is secure the vegetable/fruit demand of local members within the APCGs. Realistically, some members within an APCG may struggle producing sufficient quantity to meet their own consumption needs. Hence, food security of APCG members have been targeted.

III. Providing local food access to wider community (L2): With growing local food, APCGs can make locally grown vegetables available to the extended community such as external customers or supermarkets, greengrocers, and external consumers who are not registered with an APCG.

IV. Income and Incentive objective (I3): The "income and incentive objective" focus is to earn income and incentive from selling surplus production of APCGs to vegetable/fruit buyers and trading carbon tokens with industries.

V. Maintenance cost reduction objective (M4): This goal refers to reducing the cost of APCGs maintenance over time. For example, "maintenance cost" may represent the one time cost to build APCG platform and maintaining the database and transaction records etc. Cost related to collection and distribution of products/vegetation from members, to stores, etc. Additionally providing benefits to the members may require a payment gateway which may incur cost.

VI. Stable APCG objective (S5): The increase in the number of active APCG members, that is, those who dynamically participate in the production-sharing or carbon-sharing network, is a "stability objective".

### Part 2: Summary of variables

In order to use MCGP all variables and their deviations are identified. For APCG the idea is to identify variables and summarize their deviations to achieve ideal set of goals. The production amount and carbon tokens generated by each group will be counted as variables and maximizing/minimizing the value is considered as deviation.

### Part 3: objective classification

The objectives are classified as definite and flexible constraints based on the previous objectives (part 1). At this point, the "definite goals" are outlined as mandatory requirement on the variables, whereas the "flexible goals" are outlined as the objectives nice to have but not necessary (*Zeleny, 1976*). The classification of goals are as follows:

I. Definite goals: Maximum carbon capture objective (C). For example, the APCG's base is environmental sustainability. Thus, ecological methods must be used for APCG production.

II. Flexible goals: Goals such as local food security (F1), extended community and customer demand objective (L2), income & incentive objective (I3), maintenance cost objective (M4), and stability of APCG (S5). Refinement of these goals helps in achieving the ideal goal set, which would benefit APCG. The variables summaries is defined as: maximum C1, minimum F1, minimum L2, minimum I3, maximum M4, and minimum S5; these are termed "expected values" in the goal programming model.

### Part 4: Objective ranking

To make sure important goals met first, the priorities of the goals have been assigned. This step discusses ranking out the goals by assigning a weight (or rank) to each goal. As mentioned earlier, goals can be mutually exclusive; *i.e.,* one goal may be achievable at the expense of another. This makes it critically important to assign weights to the goals, so that least important goals are only met after the important ones. Keeping local network food security (F1) as priority, total goal set can be determined as 4!, thus in total 24 structures will be formed such as F1L2I3M4S5, F1L2M4S5I3…F1S5M4I3L2.

Part 5: Goal equation formation

Mathematical relations are developed in this section for the definite and flexible goals. Equations are as follows-

I. Carbon capture Objective (C): Organic farming methods should be used for APCG produce to increase the carbon token value.

II. Food security local demand objective (GC1): Satisfying food security of APCG should be focused. Thus, the purpose of this goal is to minimise the negative deviation from the quantity of surplus production of each APCG. Let $A_{pi}$ $E_i$ be the extra production produced by $i^{th}$ APCG, $k_0$ and $l_0$ be negative and positive variance respectively, and $t$ be the number of APCGs; then the equation for food security local demand objective (F1) would be:

$$A_{pi} \times E_i \geq 0; \forall i \leq t$$

$$A_{pi} \times E_i + k_0 - l_0 = 0; \forall i \leq t \tag{7}$$

Considering 4 APCG groups for this framework, 4 equations will be formed ($m = 4$) for each group; $N_{p1} \times E_1 + k_1 - l_1 = 0;\ \dots N_{p4} \times E_4 + k_4 - l_4 = 0;$

III. Local community demand objective (L2): The purpose of L2 is to minimise the negative variance of the total surplus production of all APCG. Assuming requirement from external supermarket is R. And positive and negative variance be s and q, respectively; then the equation will be formed as

$$\sum_{i=1}^{m} E_i \times A_{pi} \geq R$$

$$\sum_{i=1}^{m} E_i \times A_{pi} + q - s = R \tag{8}$$

IV. Income & Incentive objective (I3): Obtaining higher income is another requirement of the framework. The minimum income expectation of the ith APCG be Ii, and positive and negative variance be q1 and s1 respectively; then the equation for this objective will be minimizing negative variance

$$\sum_{i=1}^{n} I_i \times E_i \times A_{pi} \geq I$$

$$\sum_{i=1}^{n} I_i \times E_i \times A_{pi} + q1 - s1 = I \tag{9}$$

V. Maintenance cost objective (M4): Let say the maintenance cost allowances be M, and the positive and negative variance be q2 and s2, respectively; equation for the maintenance cost objective (GC4) is obtained with Equation 5.6, where Ci is the coefficient, represents the cost rate of ith APCG.

$$\sum_{i=1}^{n} C_i \times E_i \times A_{pi} \leq M$$

$$\sum_{i=1}^{n} C_i \times E_i \times A_{pi} + q2 - s2 = M \tag{10}$$

VI. Sustainability objective (GC5): Let P be the minimum number of prosumers who are participating in APCG, and positive and negative variance be q3 and s3, respectively; then, the formula for the sustainability objective (G5) would be:

$$\sum_{i=1}^{n} A_{pi} \geq P$$

$$\sum_{i=1}^{n} A_{pi} + q_3 - s_3 = P \tag{11}$$

### Part 6 Development of objective functions

Finally the objective function of each goal is formulated and, best possible solution is formed by minimizing the deviations from each goal. The objective functions here are the [(k1, k2, k3, k4), q, q₁, s₂, q₃]. *Partitioning algorithm* is used to solve this linear goal programming problem.

### Goal programming solution

As discussed previously, 24 priority goal structure sets are identified along with different ranking order. The partitioning algorithm is utilized as a solution here, in order to solve the linear goal programming problem (*Rathnayaka et al., 2015b*). The solution working principle implies on the definition of priority structures which implies that higher-order goals must be optimised before lower-order goals are even considered. The solution procedure is shown in Fig. 11 which consists of solving a series of linear programming sub-problems by using the solution of the higher-priority problem solved prior to the lower-priority problem. All the sub-problems assigned to a higher priority goals are solved first using the partitioning algorithm. The ideal tableau for this sub-problem is then examined for alternative ideal solutions. If none exists, then the present solution is ideal for the original problem with respect to all the priorities.

The algorithm then substitutes the values of the parameters for the flexible goals of the lower priorities to calculate their satisfaction levels, and the problem is solved. However, if alternative ideal solutions do exist, the next set of flexible goal and their objective function terms are added to the problem. This brings the algorithm to the next sub-problem in the series, and the optimisation resumes. The algorithm continues in this manner until no alternative ideal exists for one of the sub-problems or until all priorities have been included in the optimisation (*Ravindran, 2009*; *Zeleny, 1976*).

Goal management problem provides the best solution by comparing the achievable set of goals when compared to the predetermined goals. Additionally the identification of the necessary alterations to parameters are explained well in order to achieve all the goals in different priority structures.

## RESULTS AND DISCUSSION

In this section, simulation parameters are illustrated for the verification of the frameworks.

1)          **Framework 1 APCG definition and prerequisites.**

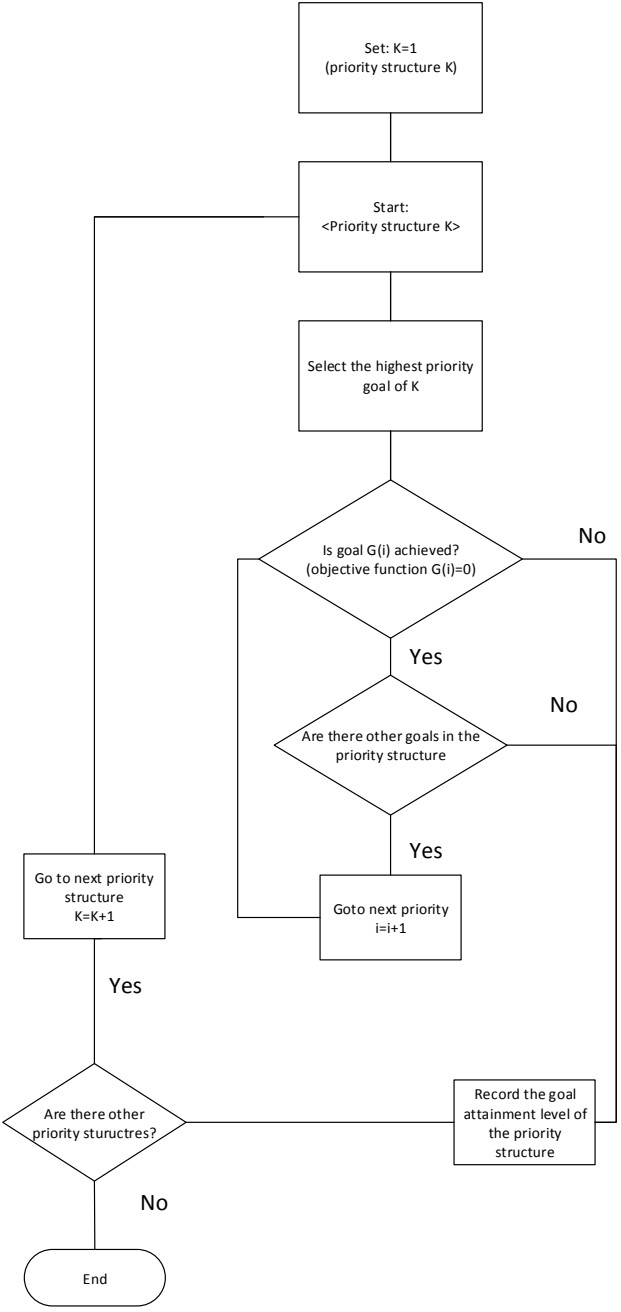

**Figure 11** **Partitioning algorithm for APCG' s goal management.**

a)   **Simulation:** As shown in Table 2, the key parameters for the verification are the prosumer production dataset. This framework is proposed using one type of crop only: lemons. It is challenging to obtain a dataset for lemon yields because prosumer community group data is not publicly available. Therefore, prosumer production profiles are generated using minimum and maximum lemon production and consumption. In the sub-section below, we discuss the generation of prosumer profile data. In this section, prosumer profiles are generated using the Australian standard production and consumption pattern (as shown in Table 3).

Country/region: In order to generate prosumer profiles, production parameters are analyzed particularly for the State of Victoria, Australia. For this study, prosumers residing in Victoria are used only to generate a sample data set. Therefore, Victorian suburban postcodes are randomly generated for prosumers. The average residential block of land is utilized to generate land sizes across Victoria. For each postcode, latitude and longitude values are determined in order to build prosumer community groups that are in close proximity.

Vegetation/fruit: Lemon trees generally produce the first crop after three years, and reach maturity when they are about five years old. Hence, the age of lemon trees and the variety are considered when estimating the minimum and maximum number of lemons produced during harvest season, and assessing the amount of carbon absorption. For this study, we consider three of the most common varieties: Eureka, Meyer and Lisbon.

Farming method: Organic and inorganic methods affect the production by 10–30%. Organic methods that involve composting, no tilling and no chemical fertilizers can reduce the quantity produced by 20–30%. Thus, this input is also considered when generating the dataset.

Lemon Consumption Rate: For prosumers, it is important to estimate their family consumption and calculate the surplus production that can be shared with the community or market. To do so, the per capita consumption of lemons is estimated and average family size is determined. Finally, prosumer consumption is calculated and averaged out to obtain the lowest production and highest production rates.

Lemon Production Rate: As a lemon tree ages, its yield increases. When it reaches maturity after five years or so, it can produce an average of ∼1500 lemons. The total amount produced also depends on whether organic or inorganic farming methods have been used. Therefore, the farming method used and the age of the lemon tree are combined to estimate the average production for a season or a whole year. Finally, the estimated average production amount is assessed and consumption is calculated to obtain the LYC and HYC. The LYC and HYC show the maximum contribution for the season that can be expected from a prosumer. After determining the production-sharing rate, we randomly generate 200 production profiles

(shown in Fig. 12), which are then used to verify the proposed framework for APCG definition and pre-condition characteristics.

b)      **Verification process:** For this verification, R software and programming language have been used. The following parameters are used for simulating the APCG definition and the prerequisites framework.

Firstly, the agro-prosumer profiles are collected and the dataset is prepared and checked for data quality. For instance, the production and consumption of agro-prosumers are analyzed and if the maximum production share is less than 50 for a season, this profile is discarded. For this framework, 300 prosumer profiles were obtained as a sample, of which five were discarded as their HYC was less than 50.

Next, the dataset consisting of prosumer profiles is partitioned according to suburb or municipal boundaries, and irrelevant profiles are removed. Of the 300 prosumer profiles, 87 prosumers belong to "G-206 clusters" and 200 prosumers belong to "G-207 clusters". The remaining eight profiles are kept in a small extra cluster as outliers.

The resulting clusters, G-206 and G-207, are obtained after removing the outliers. These clusters are further partitioned into different prosumer groups based on their production rate using the hierarchical clustering method described in section 3.5. For G-206, hierarchical clustering resulted in four clusters. Figure 13 illustrates the number of prosumers allocated to G-206 clusters where c1, c2, c3 and c4 denote four cluster groups produced by the hierarchical method. The same hierarchical clustering is done for the G-207 cluster, which resulted in eight clusters: c1 to c8 (Fig. 14).

However, as shown in Figs. 13 and 14, some clusters have a very large number of prosumers; for instance, there are more than 30 agro-prosumers in c3 of G-206, and nearly 60 in c1 of G-207. APCGs need to have a reasonable number of members in each cluster: small clusters can cause inefficiency or overheads, and large clusters can overproduce and cause storage problems or damage (such as infections) to the produce. Hence, in this scenario, the optimization of the clusters by splitting the large clusters is done in order to ensure an appropriate number of members.

In addition, Figs. 13 and 14 show clusters which are too small where the number of agro-prosumers is less than or little more than ten. For example, cluster c2 in Fig. 13 offers only 11 agro-prosumers and c8 in Fig. 14 has only eight agro-prosumers. If the APCG fails to supply an adequate amount of produce to the buyers or market, it might not enjoy good value or strong relationships in the long term and may become unsustainable. Therefore, in this scenario, adjacent prosumer clusters are merged in order to meet the amount of production required of members. For this data set, we reduce the number of clusters, merging the neighbors into one cluster. These finalized clusters constitute the APCGs.

**Table 2  Simulation parameters to verify APCGs definition and characteristics.**

| Simulation parameters | Value |
|---|---|
| Numbers of prosumer profiles | 200 |
| Minimum and Maximum threshold distance for outlier detection | 2 km–10 km |
| Minimum agro-prosumer participants in a group | 10 |
| Maximum agro-prosumer participants in a group | 50 |
| Minimum accumulated lemon production expected from each APCGs | 50 |
| Maximum accumulated lemon production expected from each APCGs | 2000 |

We optimize the originally obtained agro-prosumer clusters into an optimal number of APCGs in order to reach the maximum and minimum number of members expected in each APCG, and the minimum amount of production from each APCG. For G-206, we divide the large clusters into two APCGs by splitting the production quantity further down (we assume 10 prosumers min. and 40 prosumers max.) in each APCG, and each APCG collectively produces quantity (at least—). These finalized clusters are illustrated below in Fig. 15 for G-206 clusters. Similarly finalized clusters are produce for G-207.

Tables 4 and 5 illustrate the numerical distribution of prosumers into APCGs for G-206 and G-207 respectively. Using the distribution, similar patterns can be used to define and characterize the APCGs. Next, the pre-condition step is used to characterize the APCGs' entry requirements. Table 6 combines the average production and summarizes the pre-condition criteria for different APCGs during a season. The pre-condition criteria are provided to any interested prosumers to give them a better understanding of the entry requirements for a community-based, produce-sharing network.

2)  **Framework 2 Agro-prosumer recruitment framework.**

a)  **Simulation:** For verification and validation of the agro-prosumer recruitment framework, the solution framework is simulated using MATLAB and Excel. The setting here is a basic set-up for the examination of the proposed framework. To verify the proposed algorithm, 50 agro-prosumers production profiles were generated, assuming that these 50 agro-prosumers have shown interest in joining APCGs. For dataset generation, production behavior along with consumption patterns from framework 1 are used. Data is obtained for summer and winter seasons for four APCGs that are defined and characterized for framework 1. Four seasons are used for the evaluation period: two summers and two winters. Thus, a prosumer is evaluated over a two-year period.

**Table 3  Parameters for generation of prosumer profile.**

| Parameters | Value |
|---|---|
| Postcodes | Victorian |
| Land size | 474sm |
| Lemon varieties | Eureka, Lisbon and Meyer |
| Number of trees | 1 |
| Tree age | 3–6 years |
| Lowest production | 0–50 units |
| Highest production | 1500 units |
| Harvest season | Winter or Summer |
| Per capita consumption | 40 |
| Family size | 1–7 |
| Farming method | Organic or Inorganic |

| Variety | Farming-method | Season | Tree-age | Family size | Consumption | Postcode | LYC | HYC | longitude | latitude |
|---|---|---|---|---|---|---|---|---|---|---|
| Lemon-Lisbon | organic | June-Oct | 4 | 6 | 240 | 3143 | 0 | 360 | 145.0194 | -37.8589 |
| Lemon-Eureka | organic | June-Aug | 4 | 4 | 160 | 3055 | 80 | 440 | 144.9422 | -37.7636 |
| Lemon-Lisbon | inorganic | June-Oct | 6 | 2 | 80 | 3143 | 520 | 1520 | 145.0194 | -37.8589 |
| Lemon-Eureka | inorganic | June-Aug | 5 | 1 | 40 | 3004 | 460 | 1460 | 144.9702 | -37.8442 |
| Lemon-Eureka | inorganic | June-Aug | 5 | 3 | 120 | 3053 | 380 | 1380 | 144.9661 | -37.8036 |
| Lemon-Meyer | inorganic | all year | 5 | 4 | 160 | 3206 | 340 | 1340 | 144.9509 | -37.8465 |
| Lemon-Lisbon | inorganic | June-Oct | 6 | 1 | 40 | 3141 | 560 | 1560 | 144.9913 | -37.8407 |
| Lemon-Meyer | inorganic | all year | 4 | 5 | 200 | 3056 | 100 | 550 | 144.9601 | -37.7663 |
| Lemon-Lisbon | organic | June-Oct | 3 | 4 | 160 | 3181 | 0 | 77.6 | 144.9955 | -37.8547 |
| Lemon-Eureka | inorganic | June-Aug | 5 | 1 | 40 | 3121 | 460 | 1460 | 145.0018 | -37.8233 |
| Lemon-Eureka | organic | June-Aug | 5 | 2 | 80 | 3056 | 320 | 1120 | 144.9601 | -37.7663 |
| Lemon-Eureka | inorganic | June-Aug | 5 | 7 | 280 | 3182 | 220 | 1220 | 144.9795 | -37.8653 |
| Lemon-Meyer | organic | all year | 5 | 4 | 160 | 3181 | 240 | 1040 | 144.9955 | -37.8547 |
| Lemon-Lisbon | inorganic | June-Oct | 6 | 3 | 120 | 3141 | 480 | 1480 | 144.9913 | -37.8407 |
| Lemon-Lisbon | inorganic | June-Oct | 3 | 7 | 280 | 3141 | 0 | 17 | 144.9913 | -37.8407 |

**Figure 12  Prosumer dataset.**

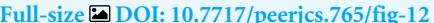

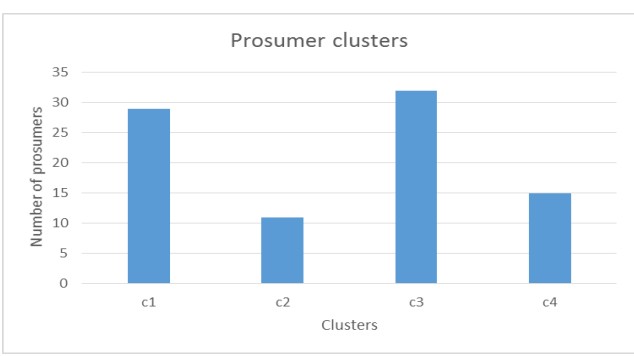

**Figure 13  Number of prosumers in each of the four clusters.**

The simulation parameters for new agro-prosumer framework are listed in Table 7. Eligible agro-prosumers are identified during the evaluation conducted after each season of the evaluation period. Only those agro-prosumers who satisfy the "eligible production

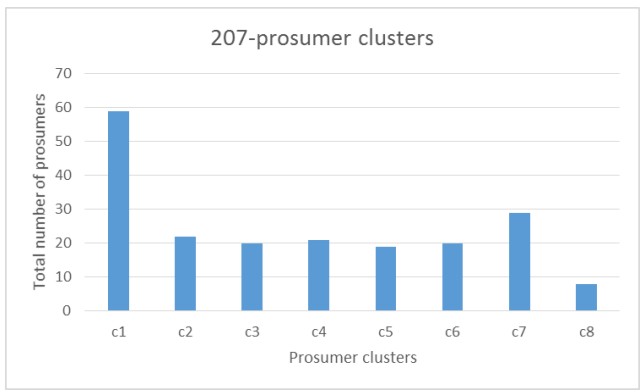

**Figure 14  Number of prosumers in G-207 cluster.**

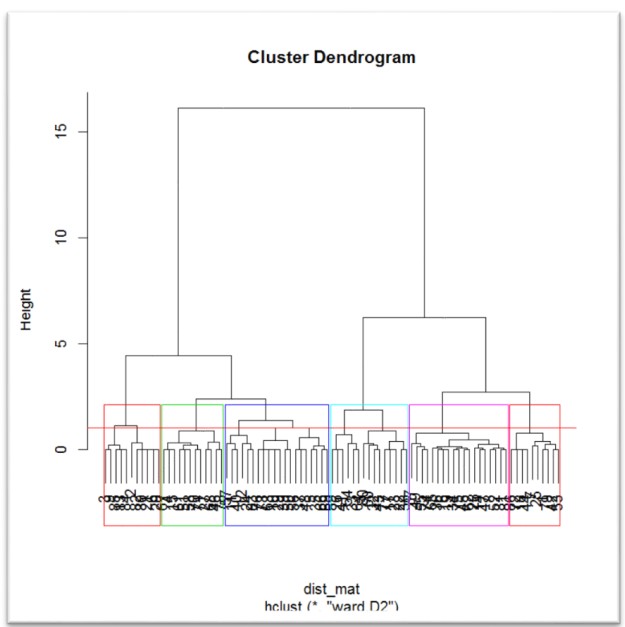

**Figure 15  APCGs for G-206.**

threshold'' in the first season can proceed to the next season. Also, eligible agro-prosumers choose their preferred APCG. The assumption here is that registered users cannot change their selection of preferred APCG until the end of the evaluation period; thus, the preferred APCG remains fixed for four seasons.

However, the eligible agro-prosumers readiness' in meeting the preferred APCG's pre-condition criteria may be irregular over the seasons during the evaluation period. To solve this issue, as we mentioned that the registered agro-prosumer is required to meet the lower threshold value of the preferred APCG to be able to meet the evaluation criteria.

**Table 4  Hierarchical clusters for G-206.**

| | | G-206 | | |
|---|---|---|---|---|
| Cluster | Total number of prosumers | LYC | HYC | Average production |
| 1 | 10 | 20 | 550 | 285 |
| 2 | 11 | 460 | 1560 | 1010 |
| 3 | 12 | 380 | 1420 | 900 |
| 4 | 20 | 220 | 1360 | 790 |
| 5 | 19 | 0 | 257 | 128.5 |
| 6 | 15 | 120 | 1000 | 560 |

**Table 5  Hierarchical cluster output for G-207.**

| | | G-207 | | |
|---|---|---|---|---|
| Cluster | Total number of prosumers | LYC | HYC | Average production |
| 1 | 59 | 0 | 257 | 128.5 |
| 2 | 41 | 320 | 1400 | 860 |
| 3 | 20 | 100 | 670 | 385 |
| 4 | 21 | 440 | 1560 | 1000 |
| 5 | 20 | 0 | 510 | 255 |
| 6 | 37 | 120 | 1260 | 690 |

**Table 6  agro-prosumer community group pre-condition criteria.**

| agro-prosumer community groups | Total number of prosumers | LYC | HYC | Average production |
|---|---|---|---|---|
| APCG1 | 59 | 0 | 257 | 128.5 |
| APCG2 | 20 | 25 | 510 | 255 |
| APCG3 | 20 | 100 | 670 | 385 |
| APCG4 | 37 | 120 | 1260 | 690 |
| APCG5 | 41 | 320 | 1400 | 860 |
| APCG6 | 21 | 440 | 1560 | 1000 |

Additionally, to determine the extent to which a registered agro-prosumer meets the pre-condition criteria of the chosen APCG, four performance indicator groups are introduced with values: "3", "2", "1" and "0" indicating "total success", "medium success", "low success" and "failure", respectively.

In this simulation, the prospect prosumers' capability in meeting their chosen APCG's pre-condition is assessed at first. Figures 16, 17, 18 and 19 show the percentage of prosumers who are allocated to different performance indices over the four seasons (or two years) for different APCGs, *i.e.,* APCG1, APCG2, APCG3 and APCG4. Result shows APCG 1 and

| Table 7 Simulation parameters. | |
|---|---|
| **Simulation parameters** | **Value** |
| Eligible production threshold(average) | 25 |
| Registered prosumers | 50 |
| Evaluation period | 2years |
| APCG1 | 0–250 |
| APCG2 | 25–550 |
| APCG3 | 100–670 |
| APCG4 | 120–1260 |

2 shows prosumers easily satisfying the pre-requisites when compared to APCG 4 which shows variation due to high entry pre-requirements.

3) **Framework 3 Goal management.**

a) **Simulation:** The solution is developed using LINGO, and is discussed in the following sub-section. Table 8 shows some of the parameters for the goal programming problem that are obtained based on the available data; some parameters are assumed based using the Australian conditions, as real data could not be accessed or found. Here, we take the four APCGs defined by APCG definition and prerequisites framework. To ease the calculations, local food security demand objective is chosen top priority and keep it the same for all the possible solution structures. Thus reducing total possible solutions to 4! *i.e.,* 24 structures. The different priority structures are formed, where the position of the characters ("F1", "L2," "I3," "M4" and "S5"] shows the priority order of the different goals. LINGO-32 is used to program the algorithm. The observations and results obtained by solving the goal problem in LINGO is presented in next section.

b) **Verification:** The solution predicts the division of the objective function according to the process priority level and the sequential solution of the resulting mixed integer linear programming model. The solution obtained at each priority level is used as a constraint at the lower level. The general examples discussed here are intended to illustrate the model's applicability to the problem of practical dimensions. For instance, I3 on priority sets the objective function for I3 to 0, but increases objective function for L2 to 35564.50. When L2 is set on priority M4 successfully met but I3 increases to 11650. When setting L2 on priority increases the I3 to 11651 and M4 to 84446. Setting M4 achieve just for M4 but does not met for L2 and I3. Same applies for S5. So, putting I3 on top achieves the most except for S5. Hence, making S5 the next priority will help to achieve all desired goals. Putting L2, I3 and M4 objective function together on same priority help achieve the best. Therefore, the negotiated priority set of goals are CF1L2I3M4S5 which is illustrated in Table 9.
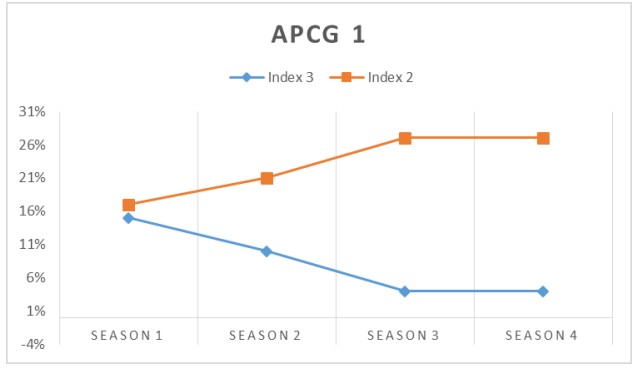

**Figure 16** APCG1 agro-prosumer percentage allocated to different performance indices over the four seasons.

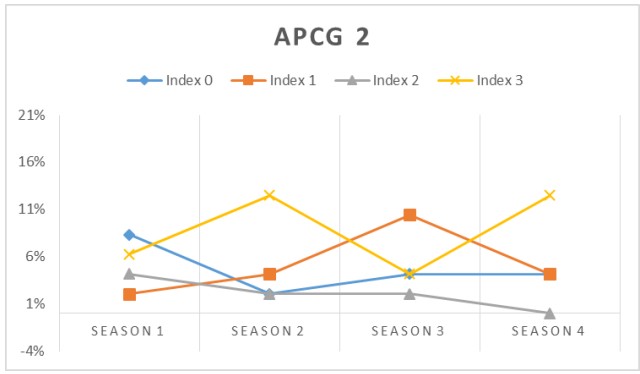

**Figure 17** APCG2 agro-prosumer percentage allocated to different performance indices over the four seasons.

## CONCLUSIONS

In order to build a seamless Agro-Prosumer Community Group structure, three key frameworks have been proposed in this paper to build a sustainable network for production sharing network. An APCG definition and prerequisites framework has been proposed to categorize the agro-prosumer profiles into feasible APCGs, while defining the pre-condition criteria for each APCG. These pre-condition criteria defined for each APCG can be utilized when recruiting new agro-prosumers, *i.e.,* the new agro-prosumers may be required to fulfil the upper and lower thresholds defined for an APCG in order to be accepted as members.

A recruitment framework is presented where, an agro-prosumer is assessed throughout the evaluation period, where his/her likelihood of meeting the APCG's pre-condition criteria and his stability is estimated, and a decision is made regarding membership of an appropriate APCG.

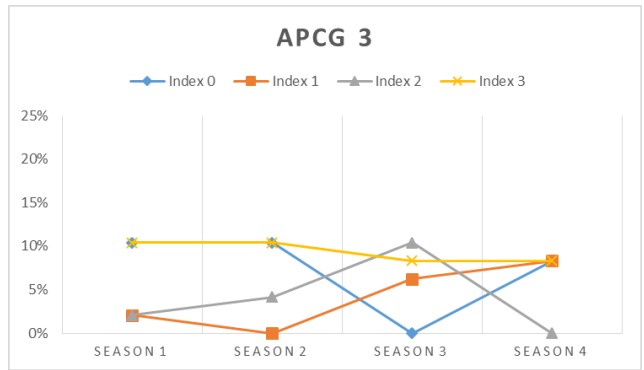

**Figure 18** APCG3 agro-prosumer percentage allocated to different performance indices over the four seasons.

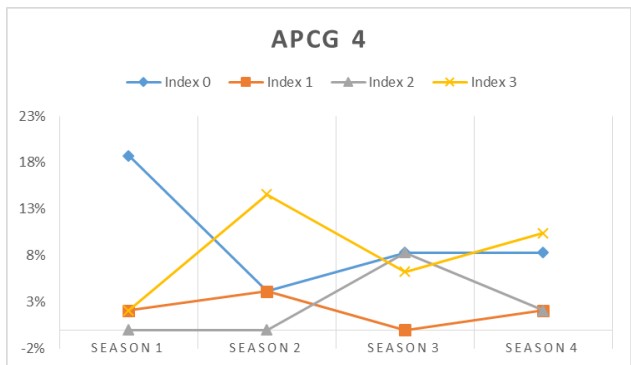

**Figure 19** APCG4 agro-prosumer percentage allocated to different performance indices over the four seasons.

Finally, a goal management framework presents an approach that determines the multiple conflicting goals within the community-based production-sharing network, prioritizes the goals based on their relative importance, and negotiates the goals to obtain the optimized set of goals for a community-based, produce-sharing network. The proposed approach for goal management assists in deciding the best priority structure. Simulation results for all three frameworks have been provided to verify the proposed framework.

## ACKNOWLEDGEMENTS

We thank the editors and the Zeleny reviewers for their kind and constructive advices.

### Funding

The authors received no funding for this work.

**Table 8   Parameters for goal programming model.**

| Simulation parameter yearly | Value |
|---|---|
| **Average production** | |
| agro-prosumer community group 1 (APCG 1) | 129 |
| APCG 2 | 255 |
| APCG 3 | 385 |
| APCG 4 | 690 |
| **Available number of prosumers** | |
| APCG 1 | 59 |
| APCG 2 | 20 |
| APCG 3 | 20 |
| APCG 4 | 37 |
| *Suburb demand (calculated for 1 suburb) | 45,000 |
| Income rate (assumed weights) APCG 1:APCG 2:APCG 3:APCG 4 | 1:3:6:9 |
| Total expected Carbon token count | 10 |
| Total expected income (assumed)** | $ 11,650 |
| Cost rate (assumed weights) APCG 1:APCG 2:APCG 3:APCG 4 | 1:2:3:4 |
| Total budgeted cost constraint (assumed)*** | $ 1,000 |
| The percentage of overall participations sustainability (Ns) | 90% |

**Table 9   Negotiated set of optimal goals.**

| Goals | Details | Value |
|---|---|---|
| GC1 | local demand of APCG | 5,440 |
| GC2 | Maintain Suburb demand | 45,000 |
| GC3 | Maximise the total expected income | $ 11,650 |
| AC | Maximise carbon token | 8 token/year |
| GC4 | Minimise the cost | $ 1,000 |
| GC5 | Sustainability | 90% |

## Competing Interests

The authors declare there are no competing interests.

## Author Contributions

- Pratima Jain performed the experiments, analyzed the data, performed the computation work, prepared figures and/or tables, and approved the final draft.
- Vidyasagr Potdar conceived and designed the experiments, authored or reviewed drafts of the paper, and approved the final draft.

## Data Availability

The raw data is available in the Tables and the code is available in the Supplemental Files.

## Supplemental Information

Supplemental information for this article can be found online at http://dx.doi.org/10.7717/peerj-cs.765#supplemental-information.

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
