# Peer review of "Frameworks for developing an Agro-Prosumer Community Group platform"

_PeerJ Computer Science, doi:10.7717/peerj-cs.765_

## Round 0.1 · original submission · Minor Revisions

The reviewers are positive about the paper but there are some places should be improved. Please provide one to one responses for the revision.

I have some comments for you to consider in preparing your revision.

The benefit of using real-time production profiles is justifiable. However, the associated cost and complexity should also be discussed and compared with those off-line versions.

Some alternative methods in addition to the hierarchical clustering method may be compared and commented.

The expressions and equations throughout the paper should be re-formulated and standardised. For example, try to use '(4)' instead of 'Equation 4'. The right-hand side brace in line 263 is not needed.

Reviewer 1 ·

Basic reporting

- The paper proposes three frameworks for Agro Prosumer Community Group.
- The first framework defines the prosumers and prerequisites. Methodology of the proposed framework is based on clustering prosumer profiles for detection of outliers and optimization of prosumer clusters.
- The novelty aspect of the paper is developing a method for recruitment of new Agro prosumer based on real-time production profile over a period of two years (evaluation period) rather than relying on historic production behavior. In this framework, authors have identified methods and developed an evaluation criteria for recruitment of eligible prosumers. Authors used MATLAB platform to test the evaluation criteria.
- In addition, authors also used a goal-oriented approach for the development of goal management framework 3 for Agro Prosumer Community Group. Authors used goal programming language for multiple conflicting goals to prioritize the goals and optimise the goal management framework using LINGO-2.

There are few suggestions:
- Equation number 3 is missing in the paper.
- There are only few references in the paper.
- In recruitment framework, authors mentioned that the recruitment of prosumers assumes that they cannot change the selection of preferred APCG. There is no discussion and hence it remains unclear how fair this assumption is.
- Also, the verification and data generation process described in framework 2 is not very clear. Authors are advised to add more details in this section.

Experimental design

- Research questions are framed properly.
- Authors have presented multiple frameworks to address the key research questions.
- Knowledge gaps are identified properly and proposed frameworks fill the gaps.

Validity of the findings

- In recruitment framework, authors mentioned that the recruitment of prosumers assumes that they cannot change the selection of preferred APCG. There is no discussion and hence it remains unclear how fair this assumption is.
- Also, the verification and data generation process described in framework 2 is not very clear. Authors are advised to add more details in this section.

·

Basic reporting

no comment

Experimental design

no comment

Validity of the findings

no comment

Additional comments

no comment

---

## Round 0.2 · accepted · Accept

The paper can be accepted. Congratulations!